# Association of electronic learning devices and online learning properties with work-related musculoskeletal disorders (WMSDs): A cross-sectional study among Thai undergraduate students

Thanyaporn Direksunthorn[1], Panicha Polpanadham[2], Ueamporn Summart[3], Khannistha Mahem[4], Pipatpong Kempanya[5], Muhamad Zulfatul A'la[6], Yuwadee Wittayapun[7,8]*

1 School of Medicine, Walailak University, Nakhon Si Thammarat, Thailand, 2 Faculty of Physical Therapy, Huachiew Chalermprakiet University, Bangkok, Thailand, 3 Faculty of Nursing, Roi Et Rajabhat University, Tha Muang, Roi Et, Thailand, 4 Boromarajonani College of Nursing, Khon Kaen, Thailand, 5 Boromarajonani College of Nursing Nakhon Phanom, Nakhon Phanom University College, Kham Thao, Thailand, 6 Faculty of Nursing, University of Jember, Jember, Indonesia, 7 School of Allied Health Sciences, Walailak University, Nakhon Si Thammarat, Thailand, 8 Movement Science and Exercise Research Center-Walailak University (MoveSE-WU), Nakhon Si Thammarat, Thailand

* yuwadee.wi@wu.ac.th

## Abstract

Computers and mobile devices are becoming the primary instruments used by students worldwide in all facets of their working and learning activities. This study aimed to investigate the relationship between the use of electronic devices, the characteristics of learning properties, and the potential predictors of work-related musculoskeletal disorders (WMSDs) among Thai undergraduate students. In this cross-sectional study, data were collected using Microsoft Forms with an online self-administered scale. The research instrument comprised four categories: demographic and health history characteristics, online learning properties, psychological health, and perceived WMSDs. Using multistage sampling, 4,618 samples were collected from 18 schools nationwide. A total of 3,705 respondents were eligible for the analysis. Descriptive statistics, chi-square, and binary logistic regression analyses were used for the data analysis. The results showed that the majority of the respondents had online learning only in some semesters/subjects (67.3%), used mobile phones for learning (43.3%), had an appropriate desk workstation (66.1%), used non-office chairs (76.0%), spent prolonged periods sitting (91.6%), had a bent posture while sitting (78.2%), had a private working space/room (92.4%), had proper lighting (85.4%), and experienced normal levels of stress (81.1%). Overall, 42.1% of Thai university students experienced WMSDs in any area of the body in the prior 6 months. Six significant predictors ($p = 0.05$) of WMSDs were obtained from the multivariate analysis, including stress, use of electronic devices, bent posture, prolonged sitting, year of study, and online learning classes (The adjusted odds ratio ranged from 1.43 to 3.67). High-risk students who mostly used mobile learning devices should be prescribed interventions to reduce stress, develop

**Data Availability Statement:** All relevant data are within the paper and its Supporting Information files.

**Funding:** The study received financial support from Walailak University through the "Walailak University's Individual Research Grants" (Grant Number WU-IRG-65-015). The funders had no role in study design, data collection and analysis, decision to publish, or preparation of the manuscript.

**Competing interests:** The authors have declared that no competing interests exist.

postural awareness and skills, emphasize positioning solutions, and reduce extended sitting time. The results indicated that preventive measures are warranted and required because the risk predictors were identified as preventable.

## Introduction

The global trend of online learning in higher education institutions has gained momentum, especially with recent advancements in technology and the COVID-19 pandemic. However, this shift towards online education has raised concerns regarding its potential impact on the musculoskeletal health of students. Prolonged sitting and improper ergonomics while using computers or digital devices have been associated with musculoskeletal disorders (MSDs) such as neck and shoulder pain, backache, and wrist strain [1–3]. A previous study revealed that the majority of computer users who spent at least 2 h per day sitting (97%), sitting with the back bent (40.8%), performing activities in a fixed position (52.1%), and taking no work breaks (95.1%) suffered from MSDs in a single region of the body (77.6%) [4]. The lack of an appropriate workspace setup, inappropriate chairs, and limited awareness of the ergonomic principles of students contribute to an increased risk of developing MSDs [5].

Musculoskeletal problems are the most common type of illness associated with work and are the primary cause of work absences or disabilities [6]. MSDs are soft tissue injuries of the muscles and tendons of the musculoskeletal system that can occur suddenly or gradually due to force, repetitive motion, vibration, and awkward posture [7]. They may affect various body parts and cover all types of illness, from minor, transient conditions to irreversible, incapacitating injuries, with substantial costs and impacts on the quality of life [3, 6, 7]. Working environments are widely acknowledged to significantly contribute to the onset and persistence of MSDs, which have been shown to have multiple etiologies [3, 6]. As such, students that participate in online learning, like other kinds of computer and mobile device users, are exposed to work risk factors daily, which can be related to some adverse physical and psychological health problems [8–10].

For undergraduate students in Thailand, classes are typically delivered in person rather than online [11]. However, before entering into the current relievingCOVID-19 situation, Thai undergraduate students faced a year of online learning [12]. Online learning requires that students spend a considerable amount of time on digital devices. Previous studies have highlighted that prolonged online learning can contribute to the development of poor physical posture among students, characterized by hunched back and bad neck postures. According to Yaseen and Salah, university students use laptops or tablets for learning for an average of 6 h per day [13]. Moreover, other studies with undergraduate students linked the result with experiencing MSDs (almost 80%) during the COVID-19 pandemic [13, 14]. They stated that taking proper precautions with postural balance might have helped the exposed students to prevent a considerable percentage of MSDs [6].

Studies have shown that students from disadvantaged backgrounds faced major postural challenges due to the unavailability of suitable space, furniture, internet connectivity, a separate room, and convenient technological devices, which compelled them to utilize electronic gadgets in poor body positions or on the floor, increasing the risk of musculoskeletal problems in the younger population [5, 15]. However, the physical learning environment of learners participating in online learning activities has rarely been investigated [16]. Additionally, previous studies have predominantly focused on identifying physical factors as primary risk factors for MSDs, neglecting to include questions pertaining to psychosocial factors [2, 17]. Furthermore,

previous studies on these problems have commonly employed a standardized Nordic questionnaire that focused on identifying the presence of musculoskeletal pain or discomfort without incorporating an additional assessment of pain intensity [2, 14]. The assessment of outcomes should incorporate the characterization of pain levels and risk perception to ensure a comprehensive approach to capturing work-related MSDs (WMSDs) [18, 19]. These gaps in the existing literature represent a significant limitation, particularly considering the crucial association between specific risk factors and the development of WMSDs among undergraduate students [13]. Hence, the primary objective of this study was to determine the prevalence rate of WMSDs among Thai undergraduate students. Additionally, we aimed to examine the comprehensive association between students' postural imbalance, electronic device usage, working environment, and psychological stress, as well as potential predictors of the specific risk factors for WMSDs. Preventing WMSDs among students is crucial because many of these disorders can be avoided. Developing an effective prevention strategy, particularly targeting conditions that contribute to pain onset in this specific population, can be obtained from the results of this study.

## Materials and methods

### Ethical approval

The study protocol adhered to the ethical guidelines and regulations of the Declaration of Helsinki. The primary ethical approval for the study protocol was obtained from the Walailak University Institutional Review Board (Ref. No. WUEC-22-007-01). The Khon Kaen University Center for Ethics in Human Research (Ref. No. HE652094) also gave its approval.

### Study design

This study was a component of a larger research project titled "Effects of e-learning during the COVID-19 pandemic on the prevalence and factors associated with musculoskeletal disorders (MSDs) among Thai, Indonesian, Vietnamese, and Laos faculty members and students." Since cross-sectional studies analyze data from a population at a point in time, this study was designed to be conducted from April to June 2022, when countrywide lockdowns and social gathering prohibitions were enacted due to the coronavirus outbreak. The study was conducted at 18 educational institutes providing bachelor's degrees in nursing, accredited by the Thailand Nursing and Midwifery Council.

### Population and sample size determination

The study population included Thai undergraduate nursing students nationwide. An infinite population proportion formula was used to obtain the sample size [20] (p = 0.70 [21], d = 0.02, and z = 1.96). A sample size of at least 2,017 was required. A low rate and inconsistent response were recognized for a sample size of 4,618 [5]. According to the simplest rules of cases-to-IVs for logistic analysis planned for use, the number of cases should be greater than 50 + 8 m, where "m" represents the number of independent variables (IVs) [22]. Thirteen IVs were used in this study. Hence, 4,618 cases exceeded the threshold of 154.

**Inclusion criteria for the sample.** Undergraduate nursing students, both female and male, aged 17–25 years and willing to participate in this study, were included as suitable participants for at least 6 months. However, female participants prevailed over male participants because of the enrollment status of females in undergraduate nursing courses in Thailand.

**Exclusion criteria for the sample.** The exclusion criteria were pregnant women; women within a year postpartum; and those with a history of kidney disease, spinal deformities, gout, rheumatoid arthritis, deformities, and back surgery.

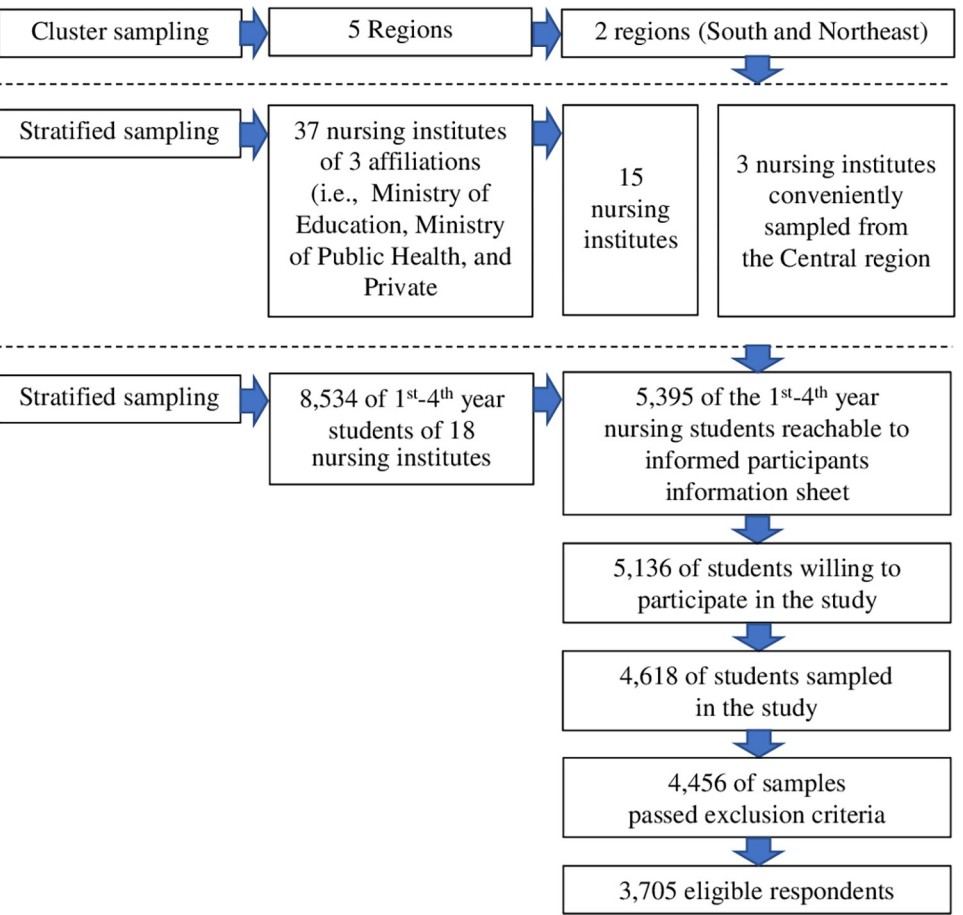

**Fig 1. Flow diagram of the sampling design through the listing of samples.**

**Sampling technique.** Based on the 2021 Thailand Nursing and Midwifery Council database [23], 96 nursing institutes were distributed across five regions. Using multistage sampling, two of the five regions were selected in the first step, namely the southern and northeastern regions. In these regions, there are 37 nursing institutes with three affiliations, including the Ministry of Education, the Ministry of Public Health, and the Private Sector. Fifteen nursing institutions were selected using a non-proportional stratified sampling technique. In addition, three nursing faculties were conveniently sampled from the central region, reaching a total of 18 faculties. The 18 institutions sampled had 8,534 students. Research partners at the 18 institutions were able to provide an information sheet and a method for providing an online voluntary consent form by clicking on a voluntary response box for 5,395 students. Of the 5,395 students, 5,136 1st–4th-year students indicated a willingness to voluntarily participate in this study. Code numbers were created to safeguard students' privacy. Of the 5,136 students, simple random sampling was performed to sample 4,618 1st–4th-year undergraduate students (**Fig 1**). Only the authors of this study had access to individual students' data.

## Measurement and data collection

**Data collection through Microsoft Forms.** The research instruments used in this study were developed based on previously validated instruments [24–27]. These instruments were adapted and designed as an online self-administered scale using Microsoft Forms (MS Forms).

The scale comprised four sections: demographic and health history characteristics, online learning properties, psychological health, and perceived WMSDs. The research instrument was considered content valid using the index of item objective congruence (IOC) by three experts, whose IOCs ranged from 0.67 to 1.00.

**Data collection through demographic and health history approach.**  For demographic information, questions on sex (male or female), age (in years), study year (1st, 2nd, 3rd, or 4th year), body weight, and height were asked. Using height and weight, the body mass index (BMI) was determined and classified into three groups: underweight ($<18.5$ kg/m$^2$), normal (18.5–22.9 kg/m$^2$), and overweight/obese ($\geq23.0$ kg/m$^2$) [28]. In terms of health history, the questionnaire included inquiries about the following conditions: pregnancy status (yes or no); being one year postpartum (yes or no); and history of operation, deformity, or disease (yes or no). Participants with a history of illness were asked to identify the specific conditions they had experienced. These conditions included kidney disease, spinal deformities, gout, rheumatoid arthritis, other deformities, and prior back surgeries.

**Data collection through online learning properties.**  The following variables related to online learning properties were assessed in this study:

1. Online learning classes: The participants were categorized based on whether they engaged in online learning for only a few semesters/subjects or the entire academic year.

2. Types of most frequently used electronic devices: The participants identified the electronic devices they most frequently used for learning activities, including mobile phones, iPads/tablets, notebooks, or personal desktop computers.

3. Appropriateness of the desk workstation: The participants assessed the suitability of their most frequently used desk workstation in terms of width, depth, and height compared to their own body, categorizing it as either appropriate or inappropriate.

4. Use of a non-office chair: The participants specified whether they used a non-office chair (e.g., backrest chair, stools, or floors) or an office chair (e.g., chair with backrest and armrests) for their learning activities.

5. Prolonged sitting: The participants indicated whether they engaged in continuous sitting for 2 h or more per day by selecting either yes or no.

6. Bent posture: The participants reported whether they maintained a bent posture continuously for 2 h or more per day, choosing either yes or no.

7. Use of a working space: The participants disclosed whether they had a designated working space, categorizing it as a private working space/room or none.

8. Perceived proper lighting: Participants assessed the adequacy of lighting in their learning environment, classifying it as either proper or improper.

**Data collection through psychological health.**  Respondents' stress levels were assessed using the Depression Anxiety Stress Scale (DASS-21). The psychometric properties of this scale have been validated across cultures [25]. To complete the scale, the participants were required to identify the symptoms they had experienced in the preceding week. Each item on the stress scale was rated on a scale of 0 (did not apply to me at all over the past week) to 3 (applied to me very much or most of the time over the past week). The stress scale consisted of seven items, yielding a total score ranging from 0 to 21. The stress levels were categorized into five groups based on their scores: "normal" (0–7), "mild" (8–9), "moderate" (10–12), "severe"

(13–16), and "extremely severe" (17+) [26]. The internal consistency of the stress scale was assessed using Cronbach's alpha coefficient. In the pilot testing phase of this study, conducted with a similar sample of 30 undergraduate students, Cronbach's alpha coefficient for the stress scale was 0.91.

**Data collection using the Nordic Musculoskeletal Questionnaire.** The Nordic Musculoskeletal Questionnaire (NMQ), which normally records MSDs in nine different body regions (i.e., feet/ankles, knees, buttocks/hips/thighs, lower back, upper back, hands/wrists, elbows, shoulders, and neck) with prevalence in the previous 7 days or 12 months [27, 29] was employed to collect data. In this study, the NMQ was expanded to encompass four additional body regions: the upper arm, forearm, fingers, and lower leg. Participants were requested to report any MSDs in any of the mentioned body regions in the past 6 months. The NMQ is a popular, accurate, and trustworthy instrument for musculoskeletal surveillance and exposure evaluation [30]. The pain level was determined using a numerical rating scale (NRS). The NRS normally consists of a list of numbers with verbal anchors ranging from 0 to 10 and indicates the full conceivable range of pain intensity [31]. The respondents were also asked whether their working or learning activities were related to musculoskeletal pain or discomfort (yes/ work-related or no). Moreover, the respondents reported WMSDs when they had MSDs with a pain level of at least 4/10 at any site and acknowledged that these pains were related to work or learning.

## Recruitment of participants

The principal researcher obtained formal authorization from the deans or directors of each nursing institute through a letter addressed to them. Additionally, the researchers established informal contacts with research collaborators responsible for data collection at each institute. The dean or director of each institute received a list of research partners associated with their respective institutes. To ensure effective communication and collaboration, the researchers conducted telephone or online meetings with 41 research colleagues from these institutes. These meetings served as a platform for discussing various aspects of the research project, including the information sheet, online informed consent process, and data collection methods. The recruitment process for student participants was conducted by research partners from the participating universities, starting from various dates between April and June 2022, which coincided with the last semester of the 2021 academic year. A total of 8,534 enrolled undergraduate students who had been studying at the participating universities for a minimum of 6 months were invited to participate in the study. The recruitment of 5,395 students was successfully carried out by representatives from each institute utilizing social media platforms, including Facebook, Line, and Zoom meetings. These channels served as effective means to reach out to potential participants and communicate the study's objectives and requirements. Students were given access to the participant information sheet and consent form through a link or QR code. All students who consented to participate in the study checked the "I accept to participate" checkbox online before the survey began. Of the 5,136 students who completed the online consent form, a simple random selection was conducted, resulting in a final sample of 4,618 students.

## Statistical analysis

Data cleaning was used to minimize statistical analysis errors (n = 4,618 respondents). This technique involved checking the response consistency and excluding cases not meeting the eligibility criteria. Descriptive statistics, such as frequencies and percentages, were used to provide an overview of the sample and variables in the study. Quantitative data were expressed as

Mean ± standard deviation. To examine for bivariate association, a chi-square test was used. The degree of the relationship was assessed using crude odds ratios (CORs) and 95% confidence intervals (Cis). The adjusted odds ratios (AORs) and 95% CIs for multivariate variables were determined using binary logistic regression analysis. Model fitting was evaluated using the Hosmer–Lemeshow goodness-of-fit test. The assumptions of the chi-squared and binary logistic regression analyses were carefully examined. All statistical analyses were performed using the Statistical Package for Social Sciences statistical software (SPSS, Version 23.0, IBM Corporation, Chicago, IL, USA).

## Results

### Characteristics of participants

This study enrolled 3,705 participants. A large proportion of participants were female (94.2%). Of the participants, 35.4% were first-year nursing students. A significant proportion of the participants were aged 20 years or older (69.3%). Approximately 51.4% of the participants had a normal BMI, with a mean BMI of 21.3 ± 3.9 kg/m$^2$. These data have been partially presented in our previous publication [32].

### Online learning risk factors and WMSDs regarding the participants (n = 3,705)

In Thailand's 2021 academic year, most respondents engaged in online learning only in some semesters or subjects (67.3%), while 43.3% used a mobile phone and 41.2% used an iPad or tablet. Some respondents reported that they had an appropriate desk workstation (66.1%) and used non-office chairs (76.0%). Most of the respondents spent prolonged periods sitting (91.6%), had a bent posture while sitting (78.2%), had private working spaces or rooms (92.4%), had proper lighting (85.4%), and experienced normal stress levels (81.1%). Over the past 6 months, 42.1% of the Thai university students experienced WMSDs in any region of their body. Two-thirds of the undergraduate students reported WMSDs at their neck (69.1%) and shoulder (62%) regions. Approximately half of the students claimed to have WMSDs, with 55.9% and 52.6% reporting that they experienced the disorders in the lower and the upper back regions, respectively (**Table 1**).

### Association between the related risk factors and WMSD prevalence

The effects of the variables on the association of WMSDs are summarized in Table 2. The relationship between the risk factors and the occurrence of musculoskeletal issues was examined using a chi-square test. Logistic regression analysis was performed to determine the association between the two variables for the CORs. The results showed that age, year of study, online learning classes, electronic devices, desk workstations, prolonged sitting, bent posture, lighting, and stress were strongly associated with WMSDs among Thai undergraduate students. The CORs of the nine positive predictors ranged from 1.27 to 4.57. The most substantial prevalence of subcategories in each significant variable associated with WMSDs was in the respondent's subgroup, including being aged between 18–19 years (49.6%), studying in the first year (49.4%), enrolling in an online learning class at 100% (51.1%), using an iPad or tablet (46.0%), adopting an inappropriate workstation (46.1%), adopting prolonged sitting periods (43.6%), sitting with a bent posture (46.9%), having improper lighting (49.6%), and experiencing extremely severe levels of stress (73.3%). Four variables showed non-significant associations with WMSDs: sex, BMI, type of chair, and working space. The findings revealed that female and male students with WMSDs comprised 42.5% and 36.3% of the study participants, respectively. Moreover, we found that university students with WMSDs who were overweight or

**Table 1. Online learning risk factors and WMSDs regarding the Thai university students (n = 3,705).**

| Category | Subcategory | N | % |
|---|---|---|---|
| **Online learning classes** | Online for only some semesters/subjects | 2,492 | 67.2 |
| | Online learning at 100% | 1,213 | 32.8 |
| **Electronic devices** | Mobile phone | 1,603 | 43.3 |
| | iPad/tablet | 1,526 | 41.2 |
| | Notebook | 526 | 14.2 |
| | Personal desktop computer | 50 | 1.3 |
| **Desk workstation** | Appropriate | 2,450 | 66.1 |
| | Inappropriate | 1,255 | 33.9 |
| **Type of chairs** | Office chair | 889 | 24.0 |
| | Non-office chair | 2,816 | 76.0 |
| | (Backrest chair) | 1,819 | 49.0 |
| | (Floor/ground) | 806 | 21.8 |
| | (Stool) | 191 | 5.2 |
| **Prolonged sitting** | Yes | 3,395 | 91.6 |
| | No | 310 | 8.4 |
| **Bent posture** | Yes | 2,898 | 78.2 |
| | No | 807 | 21.8 |
| **Working space** | Private working space/room | 3,424 | 92.4 |
| | None | 281 | 7.6 |
| **Lighting** | Proper | 3,165 | 85.4 |
| | Improper | 540 | 14.6 |
| **Stress** | Normal | 3,004 | 81.1 |
| | Mild | 319 | 8.6 |
| | Moderate | 235 | 6.3 |
| | Severe | 117 | 3.2 |
| | Extremely severe | 30 | 0.8 |
| **MSDs** | Non-cases | 1,461 | 39.4 |
| | Cases | 2,244 | 60.6 |
| **WMSDs** | Non-cases | 2,144 | 57.9 |
| | Cases | 1,561 | 42.1 |
| | (Neck) | 1,079 | 69.1 |
| | (Shoulder) | 969 | 62.1 |
| | (Lower Back) | 872 | 55.9 |
| | (Upper Back) | 821 | 52.6 |
| | (Hip/Thigh) | 320 | 20.5 |
| | (Feet/Ankle) | 320 | 20.5 |
| | (Hand) | 312 | 20.0 |
| | (Calf) | 269 | 17.2 |
| | (Upper Arm) | 164 | 10.5 |
| | (Knee) | 153 | 9.8 |
| | (Lower Arm) | 111 | 7.1 |
| | (Elbow) | 47 | 3.0 |

Abbreviations: MSDs, musculoskeletal disorder; WMSDs, work-related musculoskeletal disorder.

obese, underweight, or had a normal BMI comprised 43.3%, 43.0%, and 41.1% of the study population, respectively. The university students who had WMSDs and sat on office and non-office chairs comprised 44.4% and 41.4% of the study population, respectively. The

Table 2. Association between related risk factors and WMSDs regarding the students (n = 3,705).

| Category | Subcategory | Total | With WMSDs | | COR | (95% CI) | |
|---|---|---|---|---|---|---|---|
| | | | N | % | | Lower | Upper |
| **Gender** | Female | 3,490 | 1,483 | 42.5% | 1.30 | 0.98 | 1.73 |
| | Male | 215 | 78 | 36.3% | | | |
| **Age in year** | 18–19 | 1,137 | 564 | 49.6% | 1.55** | 1.35 | 1.79 |
| | ≥20 | 2,568 | 997 | 38.8% | | | |
| **Study year** | 1st | 1,313 | 649 | 49.4% | 1.71** | 1.41 | 2.09 |
| | 2nd | 1,011 | 391 | 38.7% | 1.11 | 0.90 | 1.36 |
| | 3rd | 767 | 298 | 38.9% | 1.11 | 0.89 | 1.39 |
| | 4th | 614 | 223 | 36.3% | | | |
| **BMI (kg/m$^2$)** | Overweight/obese (≥23.0) | 953 | 413 | 43.3% | 1.09 | 0.93 | 1.28 |
| | Underweight (<18.5) | 849 | 365 | 43.0% | 1.08 | 0.92 | 1.27 |
| | Normal (18.5–22.9) | 1,903 | 783 | 41.1% | | | |
| **Online learning class** | Online learning at 100% | 1,213 | 620 | 51.1% | 1.72** | 1.50 | 1.98 |
| | Online for only some semesters/subjects | 2,492 | 941 | 37.8% | | | |
| **Electronic devices** | iPad/tablet | 1,526 | 702 | 46.0% | 2.70* | 1.40 | 5.20 |
| | Mobile phone | 1,603 | 651 | 40.6% | 2.17* | 1.12 | 4.18 |
| | Notebook | 526 | 196 | 37.3% | 1.88 | 0.96 | 3.69 |
| | Personal desktop computer | 50 | 12 | 24.0% | | | |
| **Desk workstation** | Inappropriate | 1,255 | 578 | 46.1% | 1.27** | 1.11 | 1.46 |
| | Appropriate | 2,450 | 983 | 40.1% | | | |
| **Type of chair** | Office chair | 889 | 395 | 44.4% | 0.88 | .76 | 1.03 |
| | Non-office chair | 2,816 | 1,166 | 41.4% | | | |
| **Prolonged sitting** | Yes | 3,395 | 1,480 | 43.6% | 2.19** | 1.68 | 2.84 |
| | No | 310 | 81 | 26.1% | | | |
| **Bent posture** | Yes | 2,898 | 1,358 | 46.9% | 2.62** | 2.20 | 3.13 |
| | No | 807 | 203 | 25.2% | | | |
| **Working space** | None | 281 | 128 | 45.6% | 0.86 | .67 | 1.10 |
| | Private working space/room | 3,424 | 1,433 | 41.9% | | | |
| **Lighting** | Improper | 540 | 268 | 49.6% | 1.43** | 1.19 | 1.71 |
| | Proper | 3,165 | 1,293 | 40.9% | | | |
| **Stress (Total score = 21)** | Extremely severe (17+) | 30 | 22 | 73.3% | 4.57** | 2.03 | 10.31 |
| | Severe (13–16) | 117 | 78 | 66.7% | 3.33** | 2.25 | 4.92 |
| | Moderate (10–12) | 235 | 148 | 63.0% | 2.83** | 2.15 | 3.72 |
| | Mild (8–9) | 319 | 185 | 58.0% | 2.30** | 1.82 | 2.90 |
| | Normal (0–7) | 3,004 | 1,128 | 37.5% | | | |

Abbreviations: BMI, body mass index; CI, Confidence Interval; COR, Crude Odds Ratio; kg, kilogram; m$^2$, square meter; n, number; WMSDs, work-related musculoskeletal disorder. Asterisks indicate statistical significance, i.e.

* p < 0.05 and

** p < 0.001.

participants who had WMSDs and worked in non-private and private offices comprised 45.6% and 41.9% of the study population, respectively (**Table 2**).

## Predictors of WMSDs

The predictive factors of the different subgroups of WMSDs and their predictive values in the logistic model are presented in **Table 3**. The predictive risk factors for WMSDs among nine

**Table 3. Risk factors for predicting WMSDs in Thai undergraduate students (n = 3,705).**

| Category | Subcategory | N (Non-case) | N (Case) | Coefficient | S.E. | AOR | 95% CI | |
|---|---|---|---|---|---|---|---|---|
| | | | | | | | Lower | Upper |
| Stress | Extremely severe (17+) | 8 | 22 | 1.300 | 0.424 | 3.67* | 1.60 | 8.43 |
| | Severe (13–16) | 39 | 78 | 1.052 | 0.205 | 2.86** | 1.92 | 4.28 |
| | Moderate (10–12) | 87 | 148 | 0.939 | 0.144 | 2.56** | 1.93 | 3.39 |
| | Mild (8–9) | 134 | 185 | 0.746 | 0.123 | 2.11** | 1.66 | 2.68 |
| | Normal (0–7) | 1,876 | 1,128 | | | | | |
| Electronic devices | iPad/tablet | 824 | 702 | 1.045 | 0.350 | 2.84* | 1.43 | 5.65 |
| | Mobile phone | 952 | 651 | 0.903 | 0.350 | 2.47* | 1.24 | 4.90 |
| | Notebook | 330 | 196 | 0.760 | 0.359 | 2.14* | 1.06 | 4.32 |
| | Personal desktop computer | 38 | 12 | | | | | |
| Bent posture | Yes | 1,540 | 1,358 | 0.843 | 0.093 | 2.32** | 1.94 | 2.79 |
| | No | 604 | 203 | | | | | |
| Prolonged sitting | Yes | 1,915 | 1,480 | 0.446 | 0.142 | 1.56* | 1.18 | 2.06 |
| | No | 229 | 81 | | | | | |
| Study year | 1st | 664 | 649 | 0.435 | .0107 | 1.55** | 1.25 | 1.91 |
| | 2nd | 620 | 391 | -0.038 | 0.111 | 0.96 | 0.77 | 1.20 |
| | 3rd | 469 | 298 | 0.016 | 0.116 | 1.02 | 0.81 | 1.28 |
| | 4th | 391 | 223 | | | | | |
| Online learning class | Online learning at 100% | 593 | 620 | 0.356 | 0.076 | 1.43** | 1.23 | 1.66 |
| | Online for only some semesters/subjects | 1,551 | 941 | | | | | |
| Constant | | | | -2.776 | 0.383 | 0.062** | | |
| Percentage of correct predictions = 64.7%, chi-square of model fit = 11.732, p = 0.164 | | | | | | | | |

Abbreviations: AOR, Adjusted Odds Ratio; CI, Confidence interval; SE, Standard Error; n, number; WMSDs, work-related musculoskeletal disorders. Asterisks indicate statistical significance, i.e.

* p < 0.05 and

** p < 0.001.

significant independent variables, including stress, electronic devices, bent posture, prolonged sitting, year of study, age, online learning class, desk workstation, and lighting, were evaluated after assessing multicollinearity and multivariate variables using binary logistic regression. In the logistic regression analysis, dummy variables were created for polychotomous variables, including the study year, type of electronic device, and stress.

The results showed that six predictors, including stress, electronic devices, bent posture, prolonged sitting, study year, and online learning class, were associated with WMSDs. However, three predictors, including age, desk workstation, and lighting, were not included in the logistic model. The AOR of the six positive predictors ranged from 1.43 to 3.67. The participants in the extremely severe, severe, moderate, and mild stress group levels were 3.67, 2.86, 2.56, and 2.11 times more likely to experience WMSDs, respectively, than those who experienced normal levels of stress. The respondents who used iPads/tablets, mobile phones, and notebooks for e-learning were 2.84, 2.47, and 2.14 times more vulnerable to developing WMSDs, respectively, than personal desktop computer users, as was evident from their AOR values. Similarly, participants who adopted a bent posture and prolonged sitting were 2.32 and 1.56 times more likely to develop WMSDs, respectively, than those who adopted non-bent and non-prolonged postures. Fourth-year students were 1.55 times less likely to develop WMSDs than first-year students (AOR = 1.55, 95% CI 1.25–1.91). We found that WMSDs were more prevalent when learning was conducted entirely online than when it was conducted partially

online. The findings (constant of -2.776) showed that the data fit the logistic model
(p = 0.164), and the classification accuracy of the risk prediction model was 64.7%.

## Discussion

The rapid growth of online learning has led to a significant increase in the number of students participating in online educational activities. Understanding the specific risks and challenges faced by online students with regard to WMSDs is essential for the development of effective preventive measures and interventions. However, the incidence and prevalence of WMSDs among Thai university students are unknown. Therefore, this research investigated the primary concern of exploring the data on WMSDs in academia. Research on physical learning environments, psychosocial factors, pain intensity assessment, risk perception, and WMSDs among undergraduate students is limited in the existing literature.

According to research objectives, the findings showed that 42.1% of the participants experienced WMSDs over 6 months. Due to these linked factors, the sudden change in posture during the COVID-19 pandemic resulted in the emergence of physiological abnormalities, indicating a high risk of developing WMSDs. This is in line with what has been observed in other countries, such as Slovenia, where 39.6% [33] of university students had MSDs, and the USA, where 41% [15] of university students had MSDs. Another study conducted among Chiang Mai's smartphone-addicted students found that 30% developed MSDs [17]. This study focused on the site of pain development; the highest percentage of pain was observed in the neck, while remarkable shoulder, lower back, and upper back pain were also noted. Except for the lower back pain variation, the findings are consistent with another study on musculoskeletal disorders among students that utilize smartphones at Khon Kaen University in Thailand [34]. This variation in the percentage of musculoskeletal problems and the highest areas of complaints may be due to different predisposing factors, assessment tools, and populations.

As mentioned previously, the etiology of WMSDs is multifactorial [6, 8, 9]. The second theme this study hoped to address was the significance of online learning-related risk factors regarding WMSDs; however, causal relationships could not be identified due to the cross-sectional nature of the study. Moreover, the results of the bivariate analysis showed that sex was not significantly associated with WMSDs in the 6-month prevalence period. Due to disparities in biological and anthropometric characteristics between the sexes, the prevalence of WMSDs varies by sex [7, 21]. A similar study on musculoskeletal disorders among students has been conducted previously [14, 21].

In this study, we observed that respondents who were 18–19 years old had a 1.55 times higher possibility of developing WMSDs than those aged 20 or older. This result is consistent with a prior study that discovered the development of MSDs during the lockdown and showed statistically significant (p < 0.05) age variability [35]. Additionally, first-year students had a 1.71 times higher possibility of developing WMSDs than fourth-year students. This may be due to their lack of familiarity with university students' working/learning activities, leading to unnecessary stress that increases fatigue and decreases their body's ability to recover properly, as is evident in poor work practices [7]. These findings are consistent with a prior study conducted by Felemban et al. [36], who discovered variations in the frequency of MSDs based on the academic year, which may have been caused by various workloads.

In our study, the WMSD prevalence during the 6 months was found to be non-significantly correlated with BMI. This non-significant association between higher BMI and musculoskeletal discomfort is similar to that reported in previous studies [14, 37]. One explanation could be that the majority of the participants in the current study (51.4%) had a normal BMI. Their BMIs were not risk factors for acquiring WMSDs; therefore, more participants with higher

BMIs are needed to further investigate this association. There is controversy surrounding the relationship between BMI and MSDs because, contrary to a cross-sectional study conducted in Portugal, BMI and reported shoulder, wrist, and hand symptoms are linked to musculoskeletal discomfort [38].

Respondents who took a course entirely online during the current academic year had a 1.72 times higher risk of developing WMSDs than those who took a course partially online. This aligns with a previous study that linked the duration and degree of discomfort to the amount of time spent learning online [13]. This may also be linked to the long-term use of electronic devices during the extended period of e-learning.

Additionally, the findings indicated that respondents who used iPads/tablets and mobile phones for online learning had 2.70 and 2.17 times higher rates of developing WMSDs, respectively, than those who used personal desktop computers. This was consistent with a previous study among students who reported an association between increased MSDs and the use of desktops, laptops, or tablet computers [13]. Our investigation highlighted the highest WMSD prevalence in iPad and tablet users, which was due to the impact of long-term e-learning during the COVID-19 pandemic, eventually leading to sedentary habits in students. The findings of this study are consistent with those of earlier research on Shanghai adolescents in terms of the association between MSDs and the use of digital devices. In that study, laptop and desktop users were less likely to have MSDs [39] because the use of a personal desktop computer allows for a more flexible placement of its components (such as the screen, keyboard, or mouse) and can adopt a more natural posture and comfort, decreasing the likelihood of pain. In contrast, tablet users not only adopt a reader's posture but also frequently use one hand to touch their screens. Improper tablet use may result in inaccurate bilateral force asymmetry, leading to uneven bilateral shoulder levels. However, a tablet can be used in the same manner as smartphones [39]. Electronic devices used were consistent with a previous study, which showed that 35.1% of the participants used desktops, laptops, or tablets to study [13].

The likelihood of WMSD development correlated with an inappropriate workstation in our study, which is consistent with a previous study showing that an inappropriate workplace width was associated with a higher risk of MSD development [24]. Therefore, an individual's physical needs should be sensibly matched to befitting workstations and postures, which may be covered by an ergonomic education process.

Our results showed that chair type was not significantly associated with WMSDs. Our findings are inconsistent with those of Parvez et al. [40], who found a significant relationship between university furniture and MSD development in students. This might be because our study participants used uncomfortable non-office chairs (e.g., backrest chairs, floors, and stools) for online learning at home, which could have caused them to alter their posture frequently.

Respondents who sat for prolonged periods had a 2.19 times greater probability of developing WMSDs than those who did not. The negative effect of sitting with the respondent's back bent was more likely to cause MSDs because the adverse posture overloads the muscles and tendons surrounding the affected joints and applies excessive force on the joints. When a joint moves closest to its mid-range motion, it performs optimally. The risk of MSDs increases when joints work outside this midrange regularly or for prolonged periods without adequate recovery time [7]. This result is consistent with research from Ethiopia, which indicated that people who sat with their backs bent were four times more likely to acquire WMSDs than those who sat with their backs straight [4]. This is because poor posture can lead to stiffness and compression throughout the skeletal and muscle areas, causing discomfort and pain in numerous body parts. In accordance with another study, whether using tablets or desktop/laptop computers, students typically slouched forward when seated in chairs [13]. Therefore,

poor sitting posture throughout the study activities caused them to experience body aches [40].

The results showed that working space was not associated with WMSDs. This may explain why most participants in this study (92.4%) had private workspaces or rooms. Their working environment posed no risk of WMSD occurrence. This differs from the findings of Aschenberger et al. [16], who found that a dedicated study space was beneficial for students' motivation, focus, learning performance, and overall well-being in the classroom.

Respondents who had improper lighting had a 1.43 times greater chance of developing WMSDs than those who did not. These results are consistent with a previous study that showed that participants who had an inappropriate workstation in the context of their dimensions (width of seat and lighting intensity) had a 5.72 times greater chance of developing WMSDs [24]. According to our findings, the respondents were more likely to develop WMSDs if exposed to extremely severe, severe, moderate, or mild stress levels. This positive correlation between psychological stress and an increase in muscular tension has been postulated as a risk factor for MSDs by other researchers [37]. The final theme of interest was to assess the risk factors for WMSDs, and six significant predictors of WMSDs among Thai nursing students were identified. Since humans are multidimensional, focusing on a single cause of MSDs will limit our ability to develop a precise and accurate MSD prevention model. Evidence from electronic device users also highlights the link between work-related (physical, environmental, and psychological risk factors) and unrelated (individual risk factors) factors. Owing to the combination of personal, psychosocial, and physical risk factors, it is believed that an increase in muscle load or activity could be an early sign of musculoskeletal disorders in people who are at work [6, 9, 41]. Therefore, an assessment of multidimensional risk factors [7], such as those in the workplace (e.g., stress, bent posture, and prolonged sitting), should be a priority for the target group of first-year students with online classes and those engaging with electronic devices (e.g., iPad/tablet, mobile phone, and notebook) for online learning.

## Strength and limitations

This study had several strengths, including a large sample size and a randomly selected sample, enhancing the generalizability of the findings. The use of reliable and valid questionnaires enables comparisons between the general population and students from various disciplines. This study examined the prevalence of WMSDs among Thai university students and identified associated risk factors in the context of online learning. The major statistical analyses provided insights for a broader population, suggesting that all undergraduate students may exhibit WMSD risk factors.

However, this study had certain limitations. This study focused solely on undergraduate nursing students, which limits its generalizability to other populations. The cross-sectional design prevented the establishment of causal relationships between the risk factors and musculoskeletal discomfort owing to the absence of follow-up data. Self-reporting through questionnaires introduced a potential for recall bias. Additionally, response bias may have been present among students who did not participate in the study or were hesitant to do so during the recruitment process conducted through platforms such as Facebook, Line, or Zoom. However, it is challenging to conclusively determine the extent and direction of the bias.

## Conclusions

This study highlights that the extensive use of electronic learning devices is associated with a higher risk of WMSDs. The multivariate analysis identified six significant factors influencing the occurrence of WMSDs, particularly among first-year students who predominantly used

mobile devices for learning. Preventive measures should be implemented to reduce the negative consequences of these risk factors and prevent chronic pain and disability. These include addressing stress, promoting postural awareness, improving postural skills, emphasizing positioning strategies, and reducing prolonged sitting time. Clinical trials incorporating ergonomic and physical therapy interventions are recommended to alleviate WMSD pain. Further research is also required to understand the causes of and remedies for musculoskeletal discomfort among undergraduate students. Communication of posture issues to medical professionals and families is essential, and students should adopt the suggested postures when using mobile devices to mitigate the effects of poor posture. Corrective exercise is also important for improving postural habits. Providing ergonomic interventions, assessments, knowledge, and support can protect students from WMSDs, maintain physical fitness, and prevent chronic pain and disabilities.

## Supporting information

**S1 Data.**
(XLSX)

## Acknowledgments

The authors would like to acknowledge the cooperation of all the participating nursing institutes in providing student data. The authors express their gratitude to the research partners at each participating nursing institute for their assistance in obtaining samples and data. Further, we appreciate the participation of all nursing students in this study.

## Author Contributions

**Conceptualization:** Thanyaporn Direksunthorn, Panicha Polpanadham, Yuwadee Wittayapun.

**Data curation:** Yuwadee Wittayapun.

**Formal analysis:** Yuwadee Wittayapun.

**Funding acquisition:** Yuwadee Wittayapun.

**Investigation:** Thanyaporn Direksunthorn, Panicha Polpanadham, Ueamporn Summart, Khannistha Mahem, Pipatpong Kempanya, Yuwadee Wittayapun.

**Methodology:** Thanyaporn Direksunthorn, Panicha Polpanadham, Yuwadee Wittayapun.

**Project administration:** Yuwadee Wittayapun.

**Visualization:** Yuwadee Wittayapun.

**Writing – original draft:** Yuwadee Wittayapun.

**Writing – review & editing:** Thanyaporn Direksunthorn, Panicha Polpanadham, Ueamporn Summart, Khannistha Mahem, Pipatpong Kempanya, Muhamad Zulfatul A'la, Yuwadee Wittayapun.

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
