## [Decision Letter · Decision Letter 0]

17 Apr 2023

PONE-D-23-01570Association of electronic learning devices and online learning properties with work-related musculoskeletal disorders (WMSDs): A cross-sectional study among Thai undergraduate studentsPLOS ONE

Dear Dr. Wittayapun,

Thank you for submitting your manuscript to PLOS ONE. After careful consideration, we feel that it has merit but does not fully meet PLOS ONE’s publication criteria as it currently stands. Therefore, we invite you to submit a revised version of the manuscript that addresses the points raised during the review process.

We look forward to receiving your revised manuscript.

Kind regards,

Humayun Kabir, MSc in Epidemiology

Academic Editor

PLOS ONE

Journal Requirements:

“NO - Include this sentence at the end of your statement: The funders had no role in study design, data collection and analysis, decision to publish, or preparation of the manuscript”

Reviewers' comments:

Reviewer's Responses to Questions

**Comments to the Author**

1. Is the manuscript technically sound, and do the data support the conclusions?

Reviewer #1: Partly

Reviewer #2: Yes

2. Has the statistical analysis been performed appropriately and rigorously? 

Reviewer #1: Yes

Reviewer #2: Yes

3. Have the authors made all data underlying the findings in their manuscript fully available?

Reviewer #1: Yes

Reviewer #2: Yes

4. Is the manuscript presented in an intelligible fashion and written in standard English?

Reviewer #1: No

Reviewer #2: Yes

5. Review Comments to the Author

Reviewer #1: Review Report

Title: Association of electronic learning devices and online 3 learning properties with work-related musculoskeletal 4 disorders (WMSDs): A cross-sectional study among Thai 5 undergraduate students.

Manuscript number: PONE-D-23-01570.

Review version: RII

Review Comments

1. General Comments

Why electronic learning? Is that only during the era of COVID 19? It’s known that electronic learning induces Musculo-skeletal disorders. Hence, what is new? What is new among university students? When did actual Musculo-skeletal formation end?

Why cross sectional? Why not other design? What is the final sampling procedure used to reach the study participants?

2. Specific Comments

• All the sections of the manuscript are weak and needs strengthening.

• Try to present as per the PLOS ONE standard/ Journal format/

• The knowledge gap is described inadequately.

• The methods are also inadequately described and ensure the completeness of all its sub-sections.

• How and when was data quality assurance was ensured?

• Use of language e.g., multi-variate analysis needs revision.

• The results didn’t report confidence interval? How was the case to variable ratio calculated? Is the model fitted? How?

• Ensure the completeness of all sections E.g. The result and the discussion sections.

• The key terms are incomplete and the reference needs meticulous revision.

Regards,

Reviewer #2: The manuscript offers a interesting area of study. There are a few suggestions that are recommended

i) A professional proof reading is recommended for typos and grammatical errors

ii) the sampling technique needs justification

iii) the limitations of the study need to be clearly highlighted/acknowledged in the discussion component

6. PLOS authors have the option to publish the peer review history of their article (what does this mean?). If published, this will include your full peer review and any attached files.

Reviewer #1: No

Reviewer #2: No

<quillbot-extension-portal></quillbot-extension-portal>

---

## [Author Response · Author response to Decision Letter 0]

11 Jul 2023

Journal Requirements: Response

Author’s Response: I have made corrections to ensure that our manuscript confirms the PLOS ONE's style requirements.

Author’s Response: We already stated, “All students who consented to participate in the study checked the "I accept to participate" checkbox online before the survey started as part of the recruitment of participants. We did not include minors in this study.

“NO - Include this sentence at the end of your statement: The funders had no role in study design, data collection and analysis, decision to publish, or preparation of the manuscript”

We have already removed funding information that appeared in the Acknowledgments and Funding section and any other section of our manuscript. However, we added the information in our cover letter.

Author’s Response: We included our amended statement in our cover letter that “The study received financial support from Walailak University through the "Walailak University's Individual Research Grants" (Grant Number WU-IRG-65-015). 

Author’s Response: The funders had no role in study design, data collection and analysis, the decision to publish, or the preparation of the manuscript. (Please see the source of internal research grants via the website URL: https://riie.wu.ac.th/?page_id=5876.)

Author’s Response: None of authors received salary from funder. 

Author’s Response: Not applicable

Author’s Response: We included our amended statements within our cover letter.

Author’s Response: We have already generated our minimal data set as a supporting information file as (S1.XLSX).

Author’s Response: Thank you.

Author’s Response: We have already produced the minimal data set. (S1.XLSX)

Reviewers' comments: Reviewer #1

1. General Comments

? Why electronic learning? Is that only during the era of COVID 19? It’s known that electronic learning induces Musculo-skeletal disorders. Hence, what is new? What is new among university students? When did actual Musculo-skeletal formation end?

Author’s Response: We clarified these concerns in the introduction section as. 

Electronic learning has been introduced in the adverse situation of COVID19 lockdown to bring the students in a new-normal lifestyle and get them triggered against the challenges.

Although the impact of electronic learning in the induction of Musculo-skeletal disorders is literally known, the very particular effect in Thailand is reflected for the first time through this study. Additionally, the nature of disadvantages that cause postural disorder and eventually Musculo-skeletal disorders of the university students has also been unraveled for the Thai students by this study.

Formation of Musculo-skeletal disorders were reported be ended through the well-off economic condition, consciousness of healthy lifestyle, and the knowledge on the postural balance which together affirmed the improvement of the situation.

? Why cross sectional? Why not other design? 

Author’s Response: We appreciate the thoughtful question the reviewer. In fact, this study was a component of a larger research project titled “Effects of e-learning during the COVID-19 pandemic on the prevalence and factors associated with musculoskeletal disorders (MSDs) among Thai, Indonesian, Vietnamese, and Laos faculty members and students”. You may be agreed that cross-sectional studies analyze data from a population at a point in time, and this study was designed to conduct in a particular time-point from April to June 2022 during the countrywide lockdown and social gathering prohibition due to the coronavirus outbreak. 

Other designs such as longitudinal study or experimental study does not fit with our research problem and research question. 

What is the final sampling procedure used to reach the study participants?

Author’s Response: Very valuable question indeed. Simple Random Sampling (SRS) procedure was adopted as the final sampling procedure. 

2. Specific Comments

• All the sections of the manuscript are weak and needs strengthening.

Author’s Response: We have rigorously revised the manuscript and strengthened all possible weakness and errors throughout the manuscript.

• Try to present as per the PLOS ONE standard/ Journal format/

Author’s Response: We checked the format to make compatible with PLOS ONE.

• The knowledge gap is described inadequately.

Author’s Response: Thank you very much for your insightful comment. However, we have meticulously reviewed the manuscript and tried to enumerate the research gap of the study. The horror of the pandemic, its devastating role in deadliness, and its way out to mobilize the young generation through electronic learning were embarked on as part of the research question. However, the negative effect of positional and postural imbalance has been analyzed very critically incorporating other socioeconomic conditions as etiology. The factual differences between this COVID19 pandemic with other outbreaks have been considered. How the effect of electronic learning may be equally disrupting muscle and muscular health in emerging and reemerging infectious diseases leading to pandemics. 

The study also tried to emphasize that physical learning compared to electronic learning speculating not only the MSDs but also other probable physical and mental disorders. Additionally, psychosocial factors along with physical factors have pertained as the primary risk factors. The effect of electronic learning on pain intensity is also measured which is usually ignored in Nordic questionnaire assessment system. The existing literature failed to incorporate the characterization of pain levels and risk perception to ensure a comprehensive approach to capturing work-related MSDs (WMSDs), this research unveiled the setbacks of related research. 

• The methods are also inadequately described and ensure the completeness of all its sub-sections.

Author’s Response: Thank you so much for your observation. The authors have rigorously revised the methods to ensure the completeness of all of the subsections 

• How and when was data quality assurance was ensured?

Author’s Response: We appreciate your question. We ensured data quality through various steps, including content validity assessment, pilot testing, meetings with research partners, data cleaning, and verification of basic statistical assumptions. The details of the validation approach have been contextualized in the manuscript through: 

A. index of item objective congruence (IOC) by three experts, whose IOCs ranged from 0.67 to 1.00.

B. The internal consistency of the stress scale using Cronbach's alpha coefficient. 

C. Researchers-participants collaboration 

D. Participant’s Feedback from QR code; and

E. Data cleaning to check the response consistency and exclude cases not meeting the eligibility criteria.

‘• Use of language e.g., multi-variate analysis needs revision.

Author’s Response: We improved the quality of the text by utilizing English-language editing services from Editage (www.editage.com).

• The results didn’t report confidence interval? 

Author’s Response: We reported the confidence interval in Table 3-4.

How was the case to variable ratio calculated? 

Author’s Response: Thank you so much for your inquisition. The issue is detailed in the manuscript to highlight the importance of a higher case-to-variable ratio.

According to the simplest rules of cases-to-IVs for logistic analysis planned for use, the number of cases should be greater than 50 + 8 m, where “m” represents the number of independent variables (IVs) [22]. Thirteen IVs were used in this study. Hence, 4,618 cases exceeded the threshold of 154.

Is the model fitted? How?

Author’s Response: This is quite a befitting model and findings (constant of -2.776) showed that the data fitted with the logistic model (p = 0.164); the classification accuracy of the risk prediction model was 64.7%.

• Ensure the completeness of all sections E.g. The result and the discussion sections. • The key terms are incomplete and the reference needs meticulous revision.

Author’s Response: We carefully reviewed the revised manuscript to change the structure and clarity of the text to ensure that the data and discussion were presented clearly and objectively. We made revisions to the reference to ensure its accuracy. Nonetheless, we have rigorously revised all other sections to make each section complete and significantly meaningful. 

Reviewers' comments: Reviewer #2

i) A professional proof reading is recommended for typos and grammatical errors

Author’s Response: We improved the quality of the text with the help of professional English-language editing services - Editage (www.editage.com).

ii) the sampling technique needs justification

Author’s Response: Thank you so much for your valuable suggestion for improvement. The sampling technique is clarified as: 

Based on the 2021 Thailand Nursing and Midwifery Council database [23], 96 nursing institutes were distributed across five regions. Using multistage sampling, two of the five regions were selected in the first step, namely the southern and northeastern regions. In these regions, there are 37 nursing institutes with three affiliations, including the Ministry of Education, the Ministry of Public Health, and the Private Sector. Fifteen nursing institutions were selected using a non-proportional stratified sampling technique. In addition, three nursing faculties were conveniently sampled from the central region, reaching a total of 18 faculties. The 18 institutions sampled had 8,534 students. Research partners at the 18 institutions were able to provide an information sheet and a method for providing an online voluntary consent form by clicking on a voluntary response box for 5,395 students. Of the 5,395 students, 5,136 1st–4th-year students indicated a willingness to voluntarily participate in this study. Code numbers were created to safeguard students’ privacy. Of the 5,136 students, simple random sampling was performed to sample 4,618 1st–4th-year undergraduate students (Fig 1). Only the authors of this study had access to individual students’ data.

iii) the limitations of the study need to be clearly highlighted/acknowledged in the discussion component

Author’s Response: The limitations of the study are approached at the end of the discussion section. The prospect and thumb message of the study is clearly delineated in the light of the discussion component.

---

## [Decision Letter · Decision Letter 1]

5 Sep 2023

Association of electronic learning devices and online learning properties with work-related musculoskeletal disorders (WMSDs): A cross-sectional study among Thai undergraduate students

PONE-D-23-01570R1

Dear Dr. Wittayapun,

We’re pleased to inform you that your manuscript has been judged scientifically suitable for publication and will be formally accepted for publication once it meets all outstanding technical requirements.

Kind regards,

Humayun Kabir

Academic Editor

PLOS ONE

Additional Editor Comments (optional):

I find the manuscript acceptable in its current form, and no additional revisions are required. 

Reviewers' comments:

Reviewer's Responses to Questions

**Comments to the Author**

1. If the authors have adequately addressed your comments raised in a previous round of review and you feel that this manuscript is now acceptable for publication, you may indicate that here to bypass the “Comments to the Author” section, enter your conflict of interest statement in the “Confidential to Editor” section, and submit your "Accept" recommendation.

Reviewer #1: All comments have been addressed

Reviewer #2: All comments have been addressed

2. Is the manuscript technically sound, and do the data support the conclusions?

Reviewer #1: Partly

Reviewer #2: Yes

3. Has the statistical analysis been performed appropriately and rigorously? 

Reviewer #1: Yes

Reviewer #2: Yes

4. Have the authors made all data underlying the findings in their manuscript fully available?

Reviewer #1: Yes

Reviewer #2: Yes

5. Is the manuscript presented in an intelligible fashion and written in standard English?

Reviewer #1: Yes

Reviewer #2: Yes

6. Review Comments to the Author

Reviewer #1: 1. Still the introduction section should entail what it intended to entail and strong establishment of niche is needed.

2. The paper should be consistently from beginning to end.

3. The methods section still needs clarity and simplicity.

4. Th result and the consequen too.t sections should be appropriate brief and clear.

5.Revisit the references

Reviewer #2: Thank you for satisfactorily addressing the points highlighted in the previous review. The manuscript meets the required standards now.

7. PLOS authors have the option to publish the peer review history of their article (what does this mean?). If published, this will include your full peer review and any attached files.

Reviewer #1: No

Reviewer #2: No

---

## [Editor Report · Acceptance letter]

20 Oct 2023

PONE-D-23-01570R1 

Association of electronic learning devices and online learning properties with work-related musculoskeletal disorders (WMSDs): A cross-sectional study among Thai undergraduate students 

Dear Dr. Wittayapun:

I'm pleased to inform you that your manuscript has been deemed suitable for publication in PLOS ONE. Congratulations! Your manuscript is now with our production department. 

Kind regards, 

on behalf of

Dr. Humayun Kabir 

Academic Editor

PLOS ONE